# Comparative effects of lived experience-lecturer-guided and teacher-led classes on mental illness awareness among Japanese high school students

Shota Fujiwara[1,2]*, Fumie Yamazaki[3], Naoko Nakayama[4], Koshu Sugisaki[3]

1 Department of Home and Health sciences, Kamakura Women's University, Kanagawa, Japan,
2 Department of Health and Welfare, Graduate School of Niigata University of Health and Welfare, Niigata, Japan, 3 Department of Health and Sports, Niigata University of Health and Welfare, Niigata, Japan, 4 Department of Human and Social Services, Kanagawa University of Human Services, Kanagawa, Japan

* fujiwara@kamakura-u.ac.jp

## Abstract

Improving mental health literacy in adolescents is a global priority. The purpose of this study is to examine differences in students' attitudes across five components of mental health literacy—supportability, severity, susceptibility, recoverability, and preventability—by comparing classes led by individuals with lived experience of mental illness and textbook-based classes led by teachers. This study focuses on high school students in Japan, where educational interventions in mental health literacy remain limited. This quasi-experimental study examined changes in mental health literacy among 150 Japanese high school students (aged ≥15) following two types of instruction: one for students who received lessons from individuals with lived experience of mental illness (lecturer-guided), and another by a teacher using text-book content (non-lecturer-guided). A 24-item questionnaire measuring five mental health literacy components—supportability, severity, susceptibility, recoverability, and preventability—was administered at three time points: pre-, post-, and 3-month follow-up. Valid responses from 117 students were analyzed using factor analysis and ANOVA. Five factors were extracted with acceptable internal consistency. The lecturer-guided group showed significant post-intervention improvements in severity, susceptibility, and recoverability (p < .001), with partial retention at follow-up. The non-lecturer-guided group also showed modest gains, particularly in susceptibility. However, no significant changes were observed in supportability or preventability in either group. Lessons led by individuals with lived experience of mental illness can improve student understanding of key aspects of mental health literacy, particularly the seriousness, personal relevance, and treatability of mental illness. However, support and prevention-related beliefs may require more comprehensive or repeated interventions. These findings support the integration of lived-experience lectures into

**Data availability statement:** The dataset supporting the findings of this study has been uploaded to Zenodo (DOI: 10.5281/zeno-do.15558879) The data are under embargo and will be made publicly available upon acceptance of the manuscript for publication in PLOS ONE. All relevant data are within the manuscript and its Supporting information files.

**Funding:** Japan Society for the Promotion of Science (JP21K18536). The funders had no role in study design, data collection and analysis, decision to publish, or preparation of the manuscript.

**Competing interests:** The authors have declared that no competing interests exist.

mental health education and provide a multidimensional framework for evaluating educational outcomes.

## Introduction

Mental illness is a global health concern. According to the World Mental Health Surveys, which included 156,331 adults aged 18 years and older, the lifetime prevalence of any mental disorder was 28.6% for males and 29.8% for females. The peak risk of onset was at age 15, with median ages of onset being 19 for males and 20 for females. Major depression was one of the most prevalent disorders among both genders [1]. Other studies have also reported high rates of mental disorders among young people [2–4]. In particular, in high-income countries, the experience of adolescence has changed profoundly over the past two decades, with mental ill health increasing and now becoming common among adolescents [5]. Additionally, regional data suggest that adolescents in the Middle East, Africa, and Asia are at an even higher risk of depression [6]. The number of people affected by mental illness is expected to continue rising [7,8]. Adolescence is a critical period in shaping lifelong mental health [5,9], and these findings underscore the importance of early preventive interventions through school-based education.

Mental health literacy has gained attention as an important concept for promoting mental well-being and preventing mental illness among youth. Mental health literacy includes knowledge of how to maintain mental health, recognize symptoms and available treatments, avoid prejudice, and understand when and where to seek help [10]. Improvements in mental health literacy among adolescents have been linked to positive changes in attitudes and behaviors, and various educational interventions have reported such effects [11–14].

In Japan, a new compulsory high school health education curriculum titled "Prevention and Recovery from Mental Illness" was introduced in 2022, marking the first time in approximately 40 years that mental illness has been formally addressed in school education. The curriculum includes early recognition of symptoms, consultation with professionals, treatment and support options, recovery possibilities, and stigma reduction [15]. Although this policy initiative is commendable, implementation challenges remain. Many teachers report low mental health literacy levels and difficulty supporting students [16], and some are uncertain about how to teach mental health topics [17]. Moreover, university students preparing to become teachers also lack adequate knowledge and often hold negative images of mental illness, such as viewing it as something to be hidden or associated with crime [18].

Cancer education may serve as a reference model for addressing these educational challenges. When cancer education was introduced in junior high schools in 2022, it faced similar difficulties, such as teachers' lack of knowledge and challenges in dealing with affected families [19,20]. The Ministry of Education, Culture, Sports, Science and Technology (MEXT) emphasized the importance of utilizing external lecturers, such as healthcare professionals and cancer survivors [21]. In practice,

lectures given by cancer survivors have been shown to reduce fear, improve understanding of causes, and positively influence attitudes toward treatment, prevention, and recovery [22–26]. In line with this, MEXT also recommends the use of qualified external lecturers in mental illness education [27].

Several studies have reported school-based mental illness education in collaboration with external experts [28,29], and peer education by students of similar age and the use of professionally produced video materials have also been shown to improve knowledge [12,13]. Some studies suggest that classes taught by individuals with lived experience of mental illness contribute to reducing stigma [30]. Although a few studies have focused on high school students, most have targeted younger adolescents. For example, a study involving 12–13-year-olds reported that direct contact with individuals with mental illness did not enhance mental health literacy [31]. In addition, the effectiveness of interventions across all components of mental health literacy—such as supportability, preventability, and recoverability—has not been sufficiently examined [14,31,32].

The purpose of this study is to examine differences in students' attitudes across five components of mental health literacy—supportability, severity, susceptibility, recoverability, and preventability—by comparing classes led by individuals with lived experience of mental illness and textbook-based classes led by teachers, targeting high school students in Asia, a region where adolescents are considered at high risk for mental illness and where mental health literacy education for this age group remains insufficiently studied.

## Methods

### Participants

Participants were 150 first-grade students (aged ≥15) attending a public high school in Kanagawa, Japan. Written informed consent was obtained from the school principal. Participants and their parents were informed in writing of the study purpose and content, and that their grades would not be affected if they chose not to participate. Informed consent was obtained from the legal guardians of all participating students using a web-based opt-out system. The study protocol was approved by an appropriate ethics committee of Kamakura Women's University (no. 22015)

### Study design

A pre–post-test follow-up quasi-experimental design was used. Fig 1 illustrates the study framework. This study was conducted in four classrooms. The lecturer-guided class group participated in two sessions taught by an outside lecturer with lived experience of mental illness. The non-lecturer-guided class group participated in two sessions taught by a school-teacher using textbooks. Each session was conducted during health classes in November 2022.

The first session was a common lesson conducted for all classrooms, based on the official high school textbook *Shin Kōtō Hoken Taiiku* (*New High School Health and Physical Education*, MEXT-approved, 2022 edition, Taishukan Shoten). The lesson covered Chapter 1, "Health in Contemporary Society," Section 16, "Characteristics of Mental Disorders" (pp. 54–55). The contents included an overview of mental illness, such as brain structure, contributing factors, typical symptoms, and the process of onset and recovery. This textbook is publicly available for purchase in Japan.

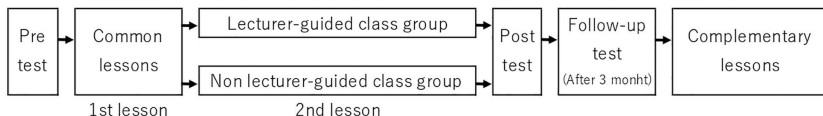

**Fig 1. Flowchart of the survey.**

The second session in the lecturer-guided group was conducted by a male lecturer with lived experience of mental illness. He developed depression at the age of 27 while working as an engineer, due to night shifts and interpersonal stress. After resigning from his job, he received treatment for approximately six months. At the time of the class, he was 30 years old, in remission, and working at a flower shop. As he did not hold a teaching license, he participated as a guest lecturer. The 50-minute class was jointly developed by the lecturer, health education teachers, and researchers, and delivered using 31 PowerPoint slides. The slides are available upon request from the corresponding author. The session followed a pre-designed narrative structure, as outlined in Table 1.

The second session in the non-lecturer-guided group was conducted by a 57-year-old male licensed health and physical education teacher. He had 32 years of teaching experience: four years at a special needs school, two years in an administrative position at the Board of Education (a role involving educational policy and teacher support rather than classroom teaching), and 26 years at a high school.The class was based on the official high school textbook *Shin Kōtō Hoken Taiiku* (*New High School Health and Physical Education*, MEXT-approved, 2022 edition, Taishukan Shoten), specifically Chapter 1, "Health in Contemporary Society," Section 17, "Responses to Mental Disorders" (pp. 56–57). The teacher utilized diagrams and explanations from the textbook, along with blackboard summaries, to deliver the content. This textbook is publicly available for purchase in Japan. The structure of the 50-minute class is shown in Table 2.

To compensate for differences in learning experiences between the lecturer-guided and non-lecturer-guided groups, a supplementary follow-up class was conducted three months later. Lecturer-guided group: Students watched a 20-minute video created by health education teachers and researchers, consisting of slides and narration. The video was based on the same textbook (*Shin Kōtō Hoken Taiiku*, 2022 edition, Taishukan Shoten) and covered equivalent content to the teacher-led session. The video source is available upon request from the corresponding author for research purposes. Non-lecturer-guided group: Students watched a 40-minute recording of the lecturer-guided class. The video was filmed from the back of the classroom to show both the lecturer and slides. Segments showing students writing reflections were removed to protect privacy. The video source is available upon request from the corresponding author for research purposes.

**Table 1. Outline of the lecturer-guided session.**

| Time | Section/ Topic | Content |
|---|---|---|
| 0–4 min | Purpose of the lecture | Explanation of the purpose of the lecture and introduction of the lecturer |
| 4–8 min | Lecturer self-introduction | Age, interests, educational background, work experience, and personality |
| 8–13 min | Process leading to onset | Work responsibilities and challenges |
| 13–18 min | Seeking medical care | Information gathering, hospital selection, second opinion, medical certificate contents, and emotions at the time of diagnosis |
| 18–25 min | Treatment to recovery | Life during treatment, interactions with others, and coping strategies for stress |
| 25–27 min | Social support | Counseling centers and telephone hotlines |

**Table 2. Outline of the non-lecturer-guided session.**

| Time | Section/ Topic | Content |
|---|---|---|
| 0–20 min | Early identification and self-care | Importance of early recognition of mental health difficulties, psychophysiological interaction, self-care strategies for stress management, and maintaining healthy sleep, diet, and physical activity |
| 20–40 min | Help-seeking behavior | Seeking support from people around, professional help-seeking, and offering supportive communication to those affected |
| 40–50 min | Realization of a mentally healthy society | Social systems, consultation services, prejudice, and discrimination |

## Measures

The questionnaire was developed based on previous studies [33–35]. It comprised 24 items across five domains: supportability, severity, susceptibility, recoverability, and preventability. One of the studies referenced also included content on the effectiveness of pharmacotherapy [33], but this was deemed inappropriate for schooling and was removed from the questionnaire. The items were reviewed, and content validity was confirmed by experts in health education and psychology, high school health and physical education teachers, and individuals with lived experience of mental illness.

The web-based survey was administered using Google Forms. The same information was used for all three surveys. The pre-survey was completed in the classroom prior to the start of class, the post-survey was completed at home after class, and the follow-up survey was completed at home approximately three months later. Responses were rated on a five-point Likert scale (1 = *strongly disagree* to 5 = *strongly agree*). Seven items were reverse coded. Data collection was from 9 November 2022–13 February 2023.

## Data analysis

Participants were 150 high school students (aged ≥ 15 years) in Japan. The questionnaire was originally developed based on previous studies. Students were surveyed three times: immediately before the lesson, immediately after the lesson, and three months later. The number of valid responses without missing data across all three surveys was 117.Responses from 117 students at three time points (N = 351) were included in the analysis. The original questionnaire consisted of 24 items. Although the items were developed with reference to previous studies, the questionnaire was newly constructed for this study and adapted for Japanese high school students. Therefore, exploratory factor analysis was conducted to examine its underlying structure in this population.

Exploratory factor analyses, including principal factor analysis and promax rotation, were conducted to identify common latent factors in the pre-, post-, and follow-up tests. In the first factor analysis, an item with a low factor loading was found to form an independent factor. Taking into consideration the factor interpretability of the item, "It is a disease caused by stress," it was excluded from the analysis. After its exclusion, factor analysis was performed again using the remaining 23 items, which resulted in a clearer factor structure and easier interpretation of each factor. Cronbach's alpha coefficients were calculated for the 23 items. A power analysis was not conducted for the present study, as the analysis was exploratory and data were collected through convenience sampling.

Differences between the mean scores of the five extracted factors were tested using a paired-sample two-way analysis of variance (ANOVA). The interaction between lecturer-guided and non-lecturer-guided groups was analysed using the Bell Curve for Excel (Social Survey Research Information Co., Ltd., Japan).

## Results

### Factor analysis

We extracted six, eight, two, four, and two items for Factors 1–5, respectively (Table 3). Factors 1–5 were termed supportability, severity, susceptibility, recoverability, and preventability, respectively. Cronbach's alpha for the 23 items was 0.79, indicating acceptable internal consistency.

### Changes post-class

The paired-sample ANOVA results showed no interaction between the lecturer-guided and non-lecturer-guided class groups (Table 4)

The ANOVA results indicated that the post-test scores for participants in the lecturer-guided class group were significantly higher than the pre-test scores for severity (4.00 vs. 4.23, respectively; $p < .001$), susceptibility (3.66 vs. 4.45, respectively; $p < .001$), and recovery (3.56 vs. 3.88, respectively; $p < .001$). Follow-up test scores for

**Table 3. Principal factor analysis with Promax rotation.**

| Item | I | II | III | IV | V |
|---|---|---|---|---|---|
| I Supportability (α = 0.893) | | | | | |
| 23. Support for those in distress is important | 0.777 | 0.082 | −0.001 | −0.040 | −0.049 |
| 24. Listening to those who are struggling is important | 0.703 | 0.067 | −0.021 | −0.018 | −0.041 |
| 22. Talking to people in distress can be preventive | 0.629 | −0.057 | −0.099 | 0.052 | 0.010 |
| 20. Talking about your problems is important | 0.621 | 0.050 | −0.082 | 0.143 | 0.049 |
| 21. This disease is preventable by relieving stress | 0.467 | 0.033 | −0.058 | −0.011 | 0.386 |
| 16. Changing one's environment can lead to recovery | 0.345 | 0.006 | 0.175 | 0.241 | 0.083 |
| II Severity (α = 0.884) | | | | | |
| * 3. This disease is not a serious problem | −0.021 | 0.671 | 0.030 | −0.081 | −0.034 |
| * 13. Young people are less affected | 0.029 | 0.571 | 0.113 | 0.096 | −0.033 |
| * 5. It is a disease to watch out for in adulthood | 0.030 | 0.541 | −0.038 | −0.058 | 0.019 |
| * 6. It is a disease that has nothing to do with me | −0.025 | 0.514 | 0.288 | 0.046 | −0.053 |
| * 18. Lifestyle is not the cause | −0.123 | 0.510 | −0.132 | 0.184 | 0.173 |
| * 4. You can go for a checkup after symptoms appear | −0.011 | 0.382 | 0.027 | 0.018 | −0.050 |
| 11. Anyone can be affected | 0.233 | 0.360 | 0.171 | −0.035 | 0.016 |
| 10. It is a life-threatening disease | 0.107 | 0.318 | 0.214 | −0.206 | 0.194 |
| III Susceptibility (α = 0.857) | | | | | |
| 1. I may be affected myself | −0.032 | 0.095 | 0.784 | 0.017 | 0.013 |
| 2. It is a disease that is familiar to me | −0.086 | 0.057 | 0.757 | 0.023 | −0.025 |
| IV Recoverability (α = 0.795) | | | | | |
| 15. Rest can lead to recovery | 0.262 | −0.082 | 0.088 | 0.614 | 0.009 |
| 14. Consultation can lead to recovery | 0.359 | −0.075 | 0.075 | 0.520 | −0.059 |
| 8. Treatment can lead to recovery | −0.038 | −0.111 | 0.006 | 0.473 | 0.229 |
| * 17. Helping others will not help you recover | 0.036 | 0.382 | −0.219 | 0.398 | −0.065 |
| V Preventability (α = 0.792) | | | | | |
| 9. This disease is preventable | −0.108 | 0.052 | 0.003 | 0.122 | 0.743 |
| 19. The disease can be prevented with daily care | 0.253 | −0.086 | 0.023 | −0.035 | 0.472 |
| 12. People affected by this disease can lead a social life | −0.159 | −0.247 | 0.029 | 0.218 | 0.075 |
| *Reversed items are negatively worded and were reverse-scored prior to analysis. | | | | | |

participants in the lecturer-guided class group were significantly higher than pre-test scores for severity (4.00 vs. 4.18, respectively; $p < .001$), susceptibility (3.66 vs. 4.24, respectively; $p < .001$), and recovery (3.56 vs. 3.72, respectively; $p < .001$), but not for supportability or preventability. Participants in the non-lecturer-guided class group had significantly higher post-test scores for susceptibility (3.44 vs. 4.31, respectively; $p < .001$) than those in the lecturer-guided class group.

## Discussion

A 24-item questionnaire was developed to investigate high school students' attitudes related to mental illness. Five factors were extracted through the factor analysis, which were consistent with the content of Japanese school education on mental illness: the importance of early recognition, the possibility of recovery through early treatment and support, and the importance of consulting specialists [15]. The results were also generally consistent with the five items of mental health literacy: support, prevention, treatment/recovery, severity, and susceptibility [33–35]. Thus, the questionnaire was considered valid.

**Table 4. Time-based changes in factor scores by group.**

| Factor | Group | Pre | | Post | | Follow-up | | Interaction | | Simple main effect | | Multiple comparison procedure (Cohen's d) |
|---|---|---|---|---|---|---|---|---|---|---|---|---|
| | | Mean | (SD) | Mean | (SD) | Mean | (SD) | F | p | F | p | |
| Supportability | Lecturer-guided class | 4.21 | (.53) | 4.27 | (.49) | 4.20 | (.67) | .45 | .64 | .78 | .459 | |
| | Non-lecturer-guided class | 4.24 | (.52) | 4.33 | (.49) | 4.17 | (.57) | | | 2.61 | .076 | post > follow-up (.33) |
| Severity | Lecturer-guided class | 4.00 | (.62) | 4.23 | (.43) | 4.18 | (.52) | 1.13 | .32 | 8.53 | <.001 | post > pre (.59), follow-up > pre (.36) |
| | Non-lecturer-guided class | 4.12 | (.48) | 4.23 | (.66) | 4.20 | (.54) | | | 1.60 | .204 | post > pre (.19) |
| Susceptibility | Lecturer-guided class | 3.66 | (.17) | 4.45 | (.17) | 4.24 | (.17) | .62 | .54 | 30.89 | <.001 | post > pre (.93), follow-up > pre (.68), post > follow-up (.35) |
| | Non-lecturer-guided class | 3.44 | (.18) | 4.31 | (.18) | 4.19 | .18) | | | 34.46 | <.001 | post > pre (.87), follow-up > pre (.84) |
| Recoverability | Lecturer-guided class | 3.56 | (.61) | 3.88 | (.58) | 3.72 | (.67) | .86 | .42 | .46 | <.001 | post > pre (.53), follow-up > pre (.23), post > follow-up (.31) |
| | Non-lecturer-guided class | 3.70 | 0.64) | 3.87 | (.65) | 3.83 | (0.59) | | | .99 | .138 | post > pre (.22) |
| Preventability | Lecturer-guided class | 3.41 | (.81) | 3.47 | (.91) | 3.46 | (.70) | .67 | .51 | .17 | .841 | |
| | Non-lecturer-guided class | 3.57 | (.89) | 3.76 | (.90) | 3.79 | (.92) | | | 2.20 | .113 | post > pre (.24), follow-up > pre (.25) |

## Severity

Severity scores significantly improved after the session in the group that was taught by the lecturer. The students likely recognized the seriousness of mental illness after hearing the lecturer's account of experiencing severe depressive symptoms that significantly impaired his daily functioning. According to the WHO, adolescence is a crucial period for shaping lifelong mental health [9]. This finding is consistent with research emphasizing the mental health burden in adolescence [4]. The present findings support this view, underscoring the importance of recognizing the severity of mental illness during this formative stage. In the field of cancer education, survivor-led classes have been shown to increase students' understanding and reduce fear by conveying the reality of illness through lived experience [22–26]. Similarly, using a lecturer with lived experience of mental illness may have made the severity of the condition more tangible and relatable for students. This approach is also consistent with MEXT's recommendation to utilize external lecturers in mental illness education [27].

## Susceptibility

Susceptibility scores significantly improved after the session in the lecturer-guided class group. The lecturer explained that he had underestimated his own risk of developing depression prior to diagnosis. He emphasized that mental illness can affect anyone, which may have contributed to students' increased awareness of their own susceptibility. This finding aligns with a previous study that reported that teaching about the side effects of pharmaceuticals, shared through teachers' experiences, can improve students' awareness of susceptibility [36]. Therefore, listening to the experiences of those who have experienced a disease/disorder in class could increase awareness of susceptibility. Asian adolescents are at particularly high risk of depression [6]. Raising awareness of this susceptibility during adolescence, as part of mental health literacy education, is vital for early prevention and timely help-seeking.

On the other hand, when teachers used textbooks in lessons, students were more aware of their own sensitivities. This may be because textbooks state that many mental illnesses begin in childhood or adolescence.

### Recoverability

Recoverability scores improved significantly in the lecturer-guided class as well. In Japan, public understanding of recovery from mental illness remains low, as noted in previous research [37]. By describing his process of receiving professional help, taking time off work, and ultimately resuming his daily and social functioning, the lecturer likely increased students' belief in the possibility of recovery. This mirrors findings from cancer education, where survivor-led sessions strengthened belief in recovery and treatment efficacy [22–26]. The lecturer's personal account may have made recovery seem more achievable and concrete to students. This suggests that narratives from people with lived experience can be a powerful tool in reducing fatalism and improving help-seeking intentions among youth.

### Supportability and preventability

In contrast, supportability and preventability did not show significant changes in the lecturer-guided group. One possible explanation is that these components are less influenced by a single narrative and may require repeated exposure to a variety of strategies or role models to produce meaningful changes. However, textbook-based instruction in the non-lecturer-guided group did produce modest improvements in these areas, possibly because it included structured guidance on communication, stress management, and early intervention. As noted in the background, teacher preparedness remains a challenge in delivering mental health content [16–17], and training teachers to effectively teach support and prevention strategies remains an important area for future development.

### Limitations and future lines of research

This study has several limitations. First, the sample consisted of four classes from a single high school, and the participants may have shared similar levels of knowledge and background characteristics, potentially introducing bias. Although the sample size was relatively small, we obtained complete data across all three time points with no missing responses, which supports the internal consistency of the findings. Moreover, the demographic characteristics of the school were similar to those of other public high schools in the region, suggesting a degree of representativeness. Nevertheless, future studies should increase the number of participating schools and students to improve generalizability and external validity.

Second, all classes were delivered by a single lecturer. As a result, the observed effects may have been influenced by the lecturer's unique teaching style, experience, or attitude. Future research should examine whether similar results are obtained when multiple lecturers are involved and explore whether common or differential effects emerge depending on the lecturer. Incorporating triangulation methods to reduce instructor-related bias is also recommended.

Third, while the exploratory factor analysis revealed a meaningful five-factor structure, we did not conduct a parallel validation using previously established scales. Further research is needed to examine the construct validity of the instrument, including both convergent and discriminant validity.

Fourth, although mental illness varies widely in type and cause across individuals, this study evaluated the effects of instruction delivered by a single individual with lived experience. Future studies should investigate whether different lecturers with diverse experiences produce consistent or varying effects.

Fifth, data were collected through an online, self-administered survey, which may be subject to response bias and data integrity issues. Alternative methods, such as paper-based or in-person data collection, should be considered to enhance data quality in future research.

In addition, incorporating qualitative data could provide deeper insights into students' experiences and perceptions, making the findings more engaging and informative for readers. Future studies should consider employing interviews or open-ended responses to complement quantitative results.

## Conclusion

The results suggest that classes taught by an outside lecturer with lived experience of mental illness can improve awareness of severity, susceptibility, and recoverability in high school mental illness education. On the other hand, the effects of the follow-up survey on perceptions of susceptibility and the possibility of recovery was smaller than that of the post-survey. Therefore, it is necessary to further consider the content of the lessons.

The low level of mental health literacy among adolescents is a common challenge in both high-income and low- and middle-income countries. Globally, school-based mental health education is receiving growing attention. This study demonstrated that classes led by individuals with lived experience of mental illness can improve key components of mental health literacy. As this approach is not culturally bound, it may be applicable to educational settings worldwide.

Moreover, by evaluating the effects of the intervention across five specific components of mental health literacy—supportability, severity, susceptibility, recoverability, and preventability—this study presents a novel framework for assessing school-based educational programs from a multidimensional perspective. These findings are expected to contribute to future international research and practice in mental health education.

## Supporting information

**S1 Dataset.  An Excel file containing anonymized data from 117 participants used in the analysis.**
(XLSX)

**S2 File  Japanese Questionnaire.** Japanese version of the questionnaire used in this study. This file contains all 24 items used in the survey.
(XLSX)

**S3 File  English Questionnaire.** English version of the questionnaire used in this study. This file contains all 24 items used in the survey.
(XLSX)

## Acknowledgments

The authors would like to thank the schools, principals, students, and legal guardians who participated in this study. We would also like to thank the teachers and lecturer who taught the classes.

## Author contributions

**Conceptualization:** Shota Fujiwara, Koshu Sugisaki.

**Data curation:** Naoko Nakayama.

**Formal analysis:** Shota Fujiwara, Fumie Yamazaki, Koshu Sugisaki.

**Funding acquisition:** Koshu Sugisaki.

**Investigation:** Shota Fujiwara.

**Methodology:** Shota Fujiwara.

**Project administration:** Shota Fujiwara.

**Writing – original draft:** Shota Fujiwara.

**Writing – review & editing:** Shota Fujiwara, Fumie Yamazaki, Naoko Nakayama, Koshu Sugisaki.

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
