## [Decision Letter · Decision Letter 0]

21 Apr 2025

PONE-D-25-03438Effectiveness of lecturer-guided vs. non-lecturer-guided classes on mental illness awareness among Japanese studentsPLOS ONE

Dear Dr. Fujiwara,

Thank you for submitting your manuscript to PLOS ONE. After careful consideration, we feel that it has merit but does not fully meet PLOS ONE’s publication criteria as it currently stands. Therefore, we invite you to submit a revised version of the manuscript that addresses the points raised during the review process.

We look forward to receiving your revised manuscript.

Kind regards,

Muhammad Zammad Aslam

Academic Editor

PLOS ONE

**Journal Requirements:**

1. When submitting your revision, we need you to address these additional requirements. Please ensure that your manuscript meets PLOS ONE's style requirements, including those for file naming. The PLOS ONE style templates can be found at https://journals.plos.org/plosone/s/file?id=wjVg/PLOSOne_formatting_sample_main_body.pdf and https://journals.plos.org/plosone/s/file?id=ba62/PLOSOne_formatting_sample_title_authors_affiliations.pdf 2. Thank you for stating the following financial disclosure: Japan Society for the Promotion of Science (JP21K18536)   Please state what role the funders took in the study.  If the funders had no role, please state: "The funders had no role in study design, data collection and analysis, decision to publish, or preparation of the manuscript." If this statement is not correct you must amend it as needed. Please include this amended Role of Funder statement in your cover letter; we will change the online submission form on your behalf. 3. We note that this data set consists of interview transcripts. Can you please confirm that all participants gave consent for interview transcript to be published? If they DID provide consent for these transcripts to be published, please also confirm that the transcripts do not contain any potentially identifying information (or let us know if the participants consented to having their personal details published and made publicly available). We consider the following details to be identifying information:- Names, nicknames, and initials- Age more specific than round numbers- GPS coordinates, physical addresses, IP addresses, email addresses- Information in small sample sizes (e.g. 40 students from X class in X year at X university)- Specific dates (e.g. visit dates, interview dates)- ID numbers Or, if the participants DID NOT provide consent for these transcripts to be published:- Provide a de-identified version of the data or excerpts of interview responses- Provide information regarding how these transcripts can be accessed by researchers who meet the criteria for access to confidential data, including:a) the grounds for restrictionb) the name of the ethics committee, Institutional Review Board, or third-party organization that is imposing sharing restrictions on the datac) a non-author, institutional point of contact that is able to field data access queries, in the interest of maintaining long-term data accessibility.d) Any relevant data set names, URLs, DOIs, etc. that an independent researcher would need in order to request your minimal data set. For further information on sharing data that contains sensitive participant information, please see: https://journals.plos.org/plosone/s/data-availability#loc-human-research-participant-data-and-other-sensitive-data If there are ethical, legal, or third-party restrictions upon your dataset, you must provide all of the following details (https://journals.plos.org/plosone/s/data-availability#loc-acceptable-data-access-restrictions):a) A complete description of the datasetb) The nature of the restrictions upon the data (ethical, legal, or owned by a third party) and the reasoning behind themc) The full name of the body imposing the restrictions upon your dataset (ethics committee, institution, data access committee, etc)d) If the data are owned by a third party, confirmation of whether the authors received any special privileges in accessing the data that other researchers would not havee) Direct, non-author contact information (preferably email) for the body imposing the restrictions upon the data, to which data access requests can be sent 4. When completing the data availability statement of the submission form, you indicated that you will make your data available on acceptance. We strongly recommend all authors decide on a data sharing plan before acceptance, as the process can be lengthy and hold up publication timelines. Please note that, though access restrictions are acceptable now, your entire data will need to be made freely accessible if your manuscript is accepted for publication. This policy applies to all data except where public deposition would breach compliance with the protocol approved by your research ethics board. If you are unable to adhere to our open data policy, please kindly revise your statement to explain your reasoning and we will seek the editor's input on an exemption. Please be assured that, once you have provided your new statement, the assessment of your exemption will not hold up the peer review process. 5. One of the noted authors is a group or consortium. In addition to naming the author group, please list the individual authors and affiliations within this group in the acknowledgments section of your manuscript. Please also indicate clearly a lead author for this group along with a contact email address. 6. Please include captions for your Supporting Information files at the end of your manuscript, and update any in-text citations to match accordingly. Please see our Supporting Information guidelines for more information: http://journals.plos.org/plosone/s/supporting-information.

Reviewers' comments:

Reviewer's Responses to Questions

**Comments to the Author**

1. Is the manuscript technically sound, and do the data support the conclusions?

Reviewer #1: Partly

Reviewer #2: Yes

Reviewer #3: Yes

2. Has the statistical analysis been performed appropriately and rigorously? 

Reviewer #1: Yes

Reviewer #2: I Don't Know

Reviewer #3: Yes

3. Have the authors made all data underlying the findings in their manuscript fully available?

Reviewer #1: No

Reviewer #2: Yes

Reviewer #3: Yes

4. Is the manuscript presented in an intelligible fashion and written in standard English?

Reviewer #1: Yes

Reviewer #2: Yes

Reviewer #3: Yes

5. Review Comments to the Author

**Reviewer #1: ** The title is suggestive but academically, it does not contain the summary of what the reader will find in the text. It is suggested that the title be reworded to more accurately reflect the content and main findings of the study, using relevant keywords that capture the essence of the research.

The abstract is attractive, but does not properly cover all the sections desirable in an academic abstract (introduction-objectives-methodology-results-conclusions). It is recommended to restructure the abstract following the IMRAD (Introduction, Methods, Results and Discussion) format, ensuring that each section is clearly represented and that the most important aspects of the research are highlighted in approximately 250-300 words.

The keywords are well chosen, but I would recommend expanding them, as far as the journal rules allow. These words, even simpler (not composed of two words or acronyms and acronyms), will help the text to be read and found in searches by academics with expertise or interest in the field. It is suggested to include between 5 and 8 keywords, prioritising simple terms and avoiding compound terms or acronyms where possible. Consider using discipline-specific thesauri to select terms that will improve the article's visibility in academic databases.

A lot of contextualisation is missing, why this article contributes something new and interesting, on a topic that has already been written about; or why what happens in this paper, specifically, is extrapolable to the whole world. It is recommended that the introduction be expanded to include a more comprehensive review of the existing literature, clearly identifying gaps in current knowledge and explaining how this study addresses them. In addition, the global relevance and applicability of the findings should be emphasised, providing concrete examples of how the work contributes to the advancement of the field.

The theoretical framework or state of the art is very weak. Some references are too old to address this issue and recent work, from 2023 and 2024, is missing. Although the texts cited are very solid and prestigious, some are somewhat ‘old’ for the area (even if they are classics) and there are few recent and international citations for a journal of this prestige and for this topic. I recommend adding 5-6 up-to-date references from top international journals. A comprehensive search of recent literature is suggested, including at least 5-6 references from the last two years (2023-2024) from high impact international journals. This will update the theoretical framework and demonstrate knowledge of the current state of research in the field.

The methodology is well validated and explained. But it should be expanded and improved, stating why the sample, although small, would be valid or extrapolated. It is recommended to include a detailed justification of the sample size and selection, explaining how representativeness and external validity of the results is ensured. Consider including a statistical power analysis or a discussion of the limitations and strengths of the chosen methodological design.

The Conclusions section is very interesting but would lack a Discussion, relating the results to the authors cited. Likewise, it would be good to include, at the end, the limitations encountered and the prospects proposed for the research community. This is a particularly valuable part of a research project on this topic. It is suggested that the Discussion section be expanded, establishing clear connections between the results obtained and the previous literature cited in the introduction. In addition, a sub-section on ‘Limitations and Future Lines of Research’ should be added, where the restrictions of the study are honestly addressed and directions for future research are proposed, highlighting how this work lays the groundwork for further advances in the field.

A harmonious distribution of paragraphs should be sought, that all have a similar length, 6-7 lines, with no long paragraphs and no loose paragraphs, as there are at the moment. This will make the text more readable and understandable. It is recommended to review the paragraph structure throughout the document, ensuring that each paragraph is of roughly similar length. Each paragraph should develop a clear main idea, with smooth transitions between them. Consider using subheadings to organise long sections and improve the navigability of the text. This restructuring will improve readability and help to keep the reader's attention and present ideas in a more coherent and organised way.

**Reviewer #2: ** Hello, dear writer, and thank you for this interesting article.

Although the topic is very interesting and interesting, I would like to offer a few tips to improve your article, which I hope will be useful and useful:

Instead of background, write the purpose and problem.

Change the keywords completely and reach five.

Reduce old references and do not lean towards before 2000.

Do not write long and short paragraphs, try to observe order.

Why do you not have a critical perspective in the background? Add it.

Your statement of the problem is a little weak. Explain in one paragraph why this research is important and necessary.

Why did you choose Japan?

Your article is international. Can you generalize the findings from Japan to some extent? Explain this a little bit on this topic as well.

Completely and several times to strengthen the structure of the journal and observe all its issues.

Summarize and critique the background of the research

Explain whether the sample is sufficient in relation to the research population and provides robust data?

I think you should strengthen the qualitative part of the article because it will provide more interesting data to the readers.

The limitations of the research are not visible in the conclusion.

Did you reach these results with only two tables? First, explain the reliability indicators, etc.

Direct quotations in the article need to be avoided a bit, although it has made the text very sweet, but it is more literary than scientific.

Unfortunately, you have not read new research in this field and this makes your article look like an old article.

Please also provide a timeline of the research process.

Best wishes

**Reviewer #3: ** This study is timely and relevant, especially in the context of Japan’s curriculum reforms that introduced mental illness education in schools. Below are strengths, concerns, and suggestions:

Strengths:

Originality: The study fills a research gap by empirically comparing lived-experience lectures with traditional textbook-based instruction.

Design: The quasi-experimental design with three assessment points is commendable.

Policy relevance: Aligns well with MEXT's guidelines and ongoing mental health education strategies.

Suggestions for Improvement:

Clarify participant number: The abstract says 177 students, but the analysis mentions 117. Please reconcile this discrepancy.

More on lecturer background: Only one lecturer was used, and his specific story may overly shape the outcomes. Future directions should discuss using multiple lecturers to validate generalizability.

Limitations:

The manuscript briefly mentions sample homogeneity (1 school); more emphasis on generalizability would strengthen the discussion.

Possible bias in self-reported data should be acknowledged.

Long-term effects: The conclusion notes no longitudinal impact despite some follow-up improvements. Clarify the criteria for this statement or rephrase it.

Formatting:

Tables need clearer headings, standard statistical notation (e.g., p < .05), and inclusion of effect sizes.

The manuscript title and short title should be reconciled (“lecturer-guided” vs. “lecturer guided”).

Expand discussion:

Consider adding comparisons to similar interventions globally, especially in cultures with stigma toward mental illness.

6. PLOS authors have the option to publish the peer review history of their article (what does this mean? ). If published, this will include your full peer review and any attached files.

**Do you want your identity to be public for this peer review?** For information about this choice, including consent withdrawal, please see our Privacy Policy .

Reviewer #1: No

Reviewer #2: **Yes: ** Amir Karimi

Reviewer #3: No

---

## [Author Response · Author response to Decision Letter 1]

5 Jul 2025

We thank the reviewers for their thoughtful comments. Please see the attached Response to Reviewers document for detailed responses to each comment.

---

## [Decision Letter · Decision Letter 1]

10 Aug 2025

PONE-D-25-03438R1Comparative Effects of Lived Experience-Lecturer-Guided and Teacher-Led Classes on Mental Illness Awareness Among Japanese High School StudentsPLOS ONE

Dear Dr. Fujiwara,

Thank you for submitting your manuscript to PLOS ONE. After careful consideration, we feel that it has merit but does not fully meet PLOS ONE’s publication criteria as it currently stands. Therefore, we invite you to submit a revised version of the manuscript that addresses the points raised during the review process.

We look forward to receiving your revised manuscript.

Kind regards,

Muhammad Zammad Aslam, Ph.D.

Academic Editor

PLOS ONE

Journal Requirements:

Additional Editor Comments:

The manuscript needs to fulfill the basic requirements following the traditional guidelines, such as explanations of teaching intervention for replication of the present study (such as detailed curriculum, description of texts or methods used, or other supporting educational material. If materials, methods, and protocols are well established, authors may cite articles where those protocols are described in detail, but the submission should include sufficient information to be understood independent of these references.

Positive Elements:

a) Evaluation of Sufficiency of Materials and Methods

The manuscript provides reasonable detail about both teaching approaches:

1. Lecturer-Guided Group; the intervention included a personal narrative from a guest lecturer with lived experience of depression. Topics covered included:

- Triggers of depression

- Recovery process

- Daily life during treatment

- Support systems

- Introduction to counselling services

- Personal message to students

Moreover, the background of the lecturer (age, past job, diagnosis timeline) is also explained in rich detail.

2. Non-Lecturer-Guided Group: A Health education teacher who is experienced in the field taught the necessary topics; Psychosomatic correlations, Seeking help, and maintaining social support

3. Curriculum & Tools:

- The five domains are assessed through a 24-item questionnaire, which is also validated by experts

- Folowchart for the Lesson plan is provided (pre/post/follow-up survey)

- Supporting materials (datasets, questionnaire in Japanese, ethics approvals, etc.) are present.

- A video follow-up (3 months later)

However, some areas should need enhancements clarifying objectively:

1. Lesson Plan Replicability:

- A full transcript may be outlined alongside the lecturer's session, alongside the narrative structure. Including an example lesson plan or structured summary would support replicability.

- Similarly, names of the chapters, names of the textbooks, or any other instructional materials may be needed alongside the teacher-led textbook content descriptions.

- The details about the videos, that is included in follow-up classes, should be needed. For instance, who is the creator? What is the script? How is this script relevant? Is the duration of the video is sufficient? What is the source of the video? etc

b. Availability of Teaching Materials:

- Please provide an English version of the questionnaire, which is a supplement for the international or multidisciplinary audiences.

- Are the lectures' notes, scripts, and slides available on request?

In summary, areas for improvement are:

- The description of lesson content, particularly for the teacher-led session, would benefit from the inclusion of exact textbook references, lesson objectives, or example discussion questions.

- For the lecturer-guided session, while the biographical narrative is informative, a more structured or script-based summary (e.g., main talking points, message themes) would be helpful for educators attempting to replicate the intervention.

- The video resources used in follow-up classes are not described in sufficient detail (e.g., content summary, creator, duration).

- The authors may consider providing an English version of the questionnaire in supplementary material for international audiences.

Educators and researchers, who are likely to adapt the intervention in different contexts, would benefit from the above changes.

Reviewers' comments:

Reviewer's Responses to Questions

**Comments to the Author**

1. If the authors have adequately addressed your comments raised in a previous round of review and you feel that this manuscript is now acceptable for publication, you may indicate that here to bypass the “Comments to the Author” section, enter your conflict of interest statement in the “Confidential to Editor” section, and submit your "Accept" recommendation.

Reviewer #1: All comments have been addressed

Reviewer #3: All comments have been addressed

2. Is the manuscript technically sound, and do the data support the conclusions?

Reviewer #1: Yes

Reviewer #3: Yes

3. Has the statistical analysis been performed appropriately and rigorously? 

Reviewer #1: (No Response)

Reviewer #3: Yes

4. Have the authors made all data underlying the findings in their manuscript fully available?

Reviewer #1: Yes

Reviewer #3: Yes

5. Is the manuscript presented in an intelligible fashion and written in standard English?

Reviewer #1: Yes

Reviewer #3: Yes

6. Review Comments to the Author

Reviewer #1: I appreciate how you have taken my comments and suggestions into account in the article, and I can clearly see all the improvements that have been implemented.

Reviewer #3: All previous reviewer comments have been adequately addressed. The authors have clarified all necessary points and incorporated feedback where applicable. No concerns regarding dual publication, research ethics, or publication ethics have arisen.

7. PLOS authors have the option to publish the peer review history of their article (what does this mean? ). If published, this will include your full peer review and any attached files.

**Do you want your identity to be public for this peer review?** For information about this choice, including consent withdrawal, please see our Privacy Policy .

Reviewer #1: No

Reviewer #3: No

---

## [Author Response · Author response to Decision Letter 2]

17 Sep 2025

Please note that we have renamed the rebuttal letter file to avoid duplicate filename issues. The latest versions have been uploaded as "Response_to_Reviewers_R2.docx," "Manuscript_R2.docx," and "Revised Manuscript with Track Changes_R2.docx." In addition, as requested, we have uploaded the English version of the questionnaire as Supporting Information labeled "S1 Questionnaire."

---

## [Editor Report · Decision Letter 2]

22 Sep 2025

Comparative Effects of Lived Experience-Lecturer-Guided and Teacher-Led Classes on Mental Illness Awareness Among Japanese High School Students

PONE-D-25-03438R2

Dear Dr. Fujiwara,

We’re pleased to inform you that your manuscript has been judged scientifically suitable for publication and will be formally accepted for publication once it meets all outstanding technical requirements.

Kind regards,

Muhammad Zammad Aslam, Ph.D.

Academic Editor

PLOS ONE
---

## [Editor Report · Acceptance letter]

PONE-D-25-03438R2

PLOS ONE

Dear Dr. Fujiwara,

I'm pleased to inform you that your manuscript has been deemed suitable for publication in PLOS ONE. Congratulations! Your manuscript is now being handed over to our production team.

Kind regards,

on behalf of

Dr. Muhammad Zammad Aslam

Academic Editor

PLOS ONE